# THE DYNAMICS OF SIGNAL PROPAGATION IN GATED RECURRENT NEURAL NETWORKS

## ABSTRACT

Training recurrent neural networks (RNNs) on long sequence tasks is plagued with difficulties arising from the exponential explosion or vanishing of signals as they propagate forward or backward through the network. Many techniques have been proposed to ameliorate these issues, including various algorithmic and architectural modifications. Two of the most successful RNN architectures, the LSTM and the GRU, do exhibit modest improvements over vanilla RNN cells, but they still suffer from instabilities when trained on very long sequences. In this work, we develop a mean field theory of signal propagation in LSTMs and GRUs that enables us to calculate the time scales for signal propagation as well as the spectral properties of the state-to-state Jacobians. By optimizing these quantities in terms of the initialization hyperparameters, we derive a novel initialization scheme that eliminates or reduces training instabilities. We demonstrate the efficacy of our initialization scheme on multiple sequence tasks, on which it enables successful training while a standard initialization either fails completely or is orders of magnitude slower. We also observe a beneficial effect on generalization performance using this new initialization.

## 1 INTRODUCTION

A common paradigm for research and development in deep learning involves the introduction of novel network architectures followed by experimental validation on a selection of tasks. While this methodology has undoubtedly generated significant advances in the field, it is hampered by the fact that the full capabilities of a candidate model may be obscured by difficulties in the training procedure. It is often possible to overcome such difficulties by carefully selecting the optimizer, batch size, learning rate schedule, initialization scheme, or other hyperparameters. However, the standard strategies for searching for good values of these hyperparameters are not guaranteed to succeed, especially if the trainable configurations are constrained to a low-dimensional subspace of hyperparameter space, which can render random search, grid search, and even Bayesian hyperparameter selection methods unsuccessful.

In this work, we argue that for long sequence tasks, the trainable configurations of *initialization hyperparameters* for LSTMs and GRUs lie in just such a subspace, which we characterize theoretically. In particular, we identify precise conditions on the hyperparameters governing the initial weight and bias distributions that are necessary to ensure trainability. These conditions derive from the observation that in order for a network to be trainable, (a) signals from the relevant parts of the input sequence must be able to propagate all the way to the loss function and (b) the gradients must be stable (i.e., they must not explode or vanish exponentially). As shown in Figure 1, training of recurrent networks with standard initialization on certain tasks begins to fail as the sequence length increases and signal propagation becomes harder to achieve. However, as we shall show, a suitably-chosen initialization scheme can dramatically improve trainability on such tasks.

We study the effect of the initialization hyperparameters on signal propagation for a very broad class of recurrent architectures, which includes as special cases many state-of-the-art RNN cells, including the GRU (Cho et al. (2014)), the LSTM (Hochreiter & Schmidhuber (1997)), and the peephole LSTM (Gers et al. (2002)). The analysis is based on the mean field theory of signal propagation developed in a line of prior work (Schoenholz et al. (2016); Xiao et al. (2018); Chen et al. (2018); Yang et al. (2019)), as well as the concept of dynamical isometry (Saxe et al. (2013); Pennington

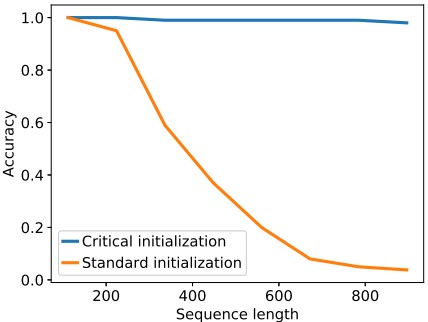

Figure 1: *Critical initialization improves trainability of recurrent networks.* Test accuracy for peephole LSTM trained to classify sequences of MNIST digits after 8000 iterations. As the sequence length increases, the network is no longer trainable with standard initialization, but still trainable using critical initialization.

et al. (2017; 2018)) that is necessary for stable gradient backpropagation and which was shown to be crucial for training simpler RNN architectures (Chen et al. (2018)). We perform a number of experiments to corroborate the results of the calculations and use them to motivate initialization schemes that outperform standard initialization approaches on a number of long sequence tasks.

## 2 BACKGROUND AND RELATED WORK

### 2.1 MEAN FIELD ANALYSIS OF NEURAL NETWORKS

Signal propagation at initialization can be controlled by varying the hyperparameters of fully-connected (Schoenholz et al. (2016); Yang & Schoenholz (2017)) and convolutional (Xiao et al. (2018)) feed-forward networks, as well as for simple gated recurrent architectures (Chen et al. (2018)). In all these cases, such control was used to obtain initialization schemes that outperformed standard initializations on benchmark tasks. In the feed-forward case, this enabled the training of very deep architectures without the use of batch normalization or skip connections.

By forward signal propagation, we specifically refer to the propagation of correlations between inputs at the start of an input sequence through the hidden states of the recurrent network at later times, as will be made precise in Section 4.2. This appears to be a necessary but not sufficient condition for trainability on certain tasks. Another is the stability of the gradients, which depend on the state-to-state Jacobian matrix, as discussed in Bengio et al. (1994). In our context, the goal of the backward propagation analysis is to improve the conditioning of the Jacobian by controlling the first two moments of its squared singular values. Forward signal propagation and the spectral properties of the Jacobian at initialization can be studied using mean field theory and random matrix theory (Poole et al. (2016); Schoenholz et al. (2016); Yang & Schoenholz (2017; 2018); Xiao et al. (2018); Pennington et al. (2017; 2018); Chen et al. (2018); Yang et al. (2019)).

As neural network training is a nonconvex problem, using a modified initialization scheme could lead to convergence to different points in parameter space in a way that adversely affects the generalization error. We provide some empirical evidence that this does not occur, and in fact, the use of initialization schemes satisfying these conditions has a beneficial effect on the generalization error.

### 2.2 THE EXPLODING/VANISHING GRADIENT PROBLEM AND SIGNAL PROPAGATION IN RECURRENT NETWORKS

The exploding/vanishing gradient problem is a well-known phenomenon that hampers training on long time sequence tasks (Bengio et al. (1994); Pascanu et al. (2013)). Apart from the gating mechanism, there have been numerous proposals to alleviate the vanishing gradient problem by constraining the weight matrices to be exactly or approximately orthogonal (Pascanu et al. (2013); Wisdom et al. (2016); Vorontsov et al. (2017); Jose et al. (2017)), or more recently by modifying some terms in the gradient (Arpit et al. (2018)), while exploding gradients can be handled by clipping (Pascanu et al. (2013)). Another recently proposed approach to ensuring signal propagation in long sequence tasks introduces auxiliary loss functions (Trinh et al. (2018)). This modification of the loss can be seen as a form of regularization. Chang et al. (2019) study the connections between recurrent networks and certain ordinary differential equations and propose the AntisymmetricRNN that can capture long term dependencies in the inputs. While many of these approaches have been quite successful, they

typically require modifying the training algorithm, the loss function, or the architecture, and as such exist as complementary methods to the one we investigate here. We postpone the investigation of a combination of techniques to future work.

### 2.3 Dynamics beyond initialization

While the analysis below applies to the network at initialization, it was recently shown that as the network width becomes large, parameters move very little from their initial values (Jacot et al. (2018); Lee et al. (2019)). A non-trivial and surprising consequence is that covariance matrices like those studied here remain constant during training. While Jacot et al. (2018); Lee et al. (2019) do not specifically study RNNs, the analysis of Yang (2019) suggests the conclusions are likely to carry over. Moreover, even for finite-width networks, simply satisfying the conditions derived in this work at initialization can lead to a dramatic improvement in trainability as we show below.

## 3 Notation

We denote matrices by bold upper case Latin characters and vectors by bold lower case Latin characters. $\mathcal{D}x$ denotes a standard Gaussian measure. The normalized trace of a random $N \times N$ matrix $\mathbf{A}$, $\frac{1}{N}\mathbb{E}\mathrm{tr}(\mathbf{A})$, is denoted by $\tau(\mathbf{A})$. $\circ$ is the Hadamard product, $\sigma(\cdot)$ is a sigmoid function and both $\sigma(\cdot), \tanh(\cdot)$ act element-wise. We denote by $\mathbf{D_a}$ a diagonal matrix with $\mathbf{a}$ on the diagonal.

## 4 Mean field analysis of signal propagation and dynamical isometry

### 4.1 Model description and important assumptions

| | Vanilla RNN | GRU Cho et al. (2014) | LSTM Hochreiter & Schmidhuber (1997) |
|---|---|---|---|
| $K$ | $\{f\}$ | $\{f, r\}$ | $\{i, f, r, o\}$ |
| $\mathbf{g}_k$ | $\cdot$ | $\sigma(\mathbf{u}_r)$ | $\cdot$ |
| $\mathbf{f}$ | $\sigma(\mathbf{u}_f^t)$ | $\sigma(\mathbf{u}_f^t) \circ \mathbf{s}^{t-1}$ $+(\mathbf{1} - \sigma(\mathbf{u}_f^t)) \circ \tanh(\mathbf{u}_{r_2}^t)$ | $\sigma(\mathbf{u}_o^t) \circ \tanh(\mathbf{c}^0+$ $\sum_{j=1}^{t} \prod_{l=j}^{t} \sigma(\mathbf{u}_f^l) \circ \sigma(\mathbf{u}_i^j) \circ \tanh(\mathbf{u}_r^j))$ |

Table 1: A number of recurrent architectures written in the form 1. $K$ is the set of pre-activation subscripts, $f$ is the state update function in eqn. (1a) and $\mathbf{g}_k$ is the function in eqn. (1c). The LSTM cell state is unrolled in order to emphasize that it can be written as a function of variables that are Gaussian at the large $N$ limit. See Table 3 for additional architectures.

We begin with a general description of recurrent architectures that can be specialized to the GRU, LSTM and peephole LSTM among others. We denote the state of a recurrent network at time $t$ by $\mathbf{s}^t \in \mathbb{R}^N$ with $s_i^0 \sim \mathcal{D}_0$, and a sequence of inputs to the network by $\{\mathbf{z}^1, ..., \mathbf{z}^T\}, \mathbf{z}^t \in \mathbb{R}^N$.

The state evolution of the network is given by

$$\mathbf{s}^t = \mathbf{f}(\mathbf{s}^{t-1}, \{\mathbf{u}_k^1\}, ..., \{\mathbf{u}_k^t\}, \mathbf{z}^t) \tag{1a}$$

where $\mathbf{f}$ is an element-wise, affine function of $\mathbf{s}^{t-1}$ and $\{\mathbf{u}_k^t\}$ is a set of pre-activations $\mathbf{u}_k^t \in \mathbb{R}^N, k \in K$ defined for a set of subscripts $K$. They are given by

$$\mathbf{u}_k^t = \mathbf{W}_k \mathbf{s}^{t-1} + \mathbf{U}_k \mathbf{z}^t + \mathbf{b}_k \tag{1b}$$

where $\mathbf{W}_k, \mathbf{U}_k \in \mathbb{R}^{N \times N}, \mathbf{b}_k \in \mathbb{R}^N$. We define additional pre-activations

$$\mathbf{u}_{k_2}^t = \mathbf{W}_{k_2} \mathbf{D}_{\mathbf{g}_k(\mathbf{u}_k^t)} \mathbf{s}^{t-1} + \mathbf{U}_{k_2} \mathbf{z}^t + \mathbf{b}_{k_2} \tag{1c}$$

where $\mathbf{g}_k : \mathbb{R}^N \to \mathbb{R}^N$ is an element-wise function and $\mathbf{u}_k^t$ is defined as in eqn. (1b) [1]. In cases where there is no need to distinguish between variables of the form (1b) and (1c) we will refer to both as

---

[1] Variables of the form (1c) will be present only in the GRU (see Table 1).

$\mathbf{u}_k^t$. This state update equation describes all the architectures studied in this paper, as well as many others, as detailed in Tables 1,3. The pre-activations $\mathbf{u}_k^t$, which are typically used to construct gates in recurrent network with the application of appropriate nonlinearities, will be of interest because they will tend in distribution to Gaussians as the network width is taken to infinity.

We assume $W_{k,ij} \sim \mathcal{N}(0, \sigma_k^2/N), U_{k,ij} \sim \mathcal{N}(0, \nu_k^2/N), b_{k,i} \sim \mathcal{N}(\mu_k, \rho_k^2)$ i.i.d. and denote $\Theta = \bigcup_k \{\sigma_k^2, \nu_k^2, \rho_k^2, \mu_k\}$. As in Chen et al. (2018), we make the *untied weights assumption* that $\mathbf{W}_k$ is independent of $\mathbf{s}^t$. Although the assumption of untied weights may seem counter-intuitive in the context of recurrent networks, recall that we are only interested in characterizing various statistics of the network, not in computing the entire input-output map, for which using tied weights would certainly be crucial. For merely characterizing certain statistics, the untied weights assumption actually has long history of yielding correct predictions, and therefore it is not at all unnatural to adopt: 1) when computing the covariance of gradients in an MLP, the weights during backpropagation can be treated as independently sampled from the weights used during the forward propagation (Schoenholz et al., 2016; Yang & Schoenholz, 2017); 2) when computing the Jacobian singular value distribution of an MLP, the pre-activations $\mathbf{h} = \mathbf{W}\mathbf{x}$ can be treated as the result of multiplying an iid copy $\mathbf{W}'$ of $\mathbf{W}$ (i.e. $\mathbf{h} = \mathbf{W}'\mathbf{x}$) (Pennington et al., 2017; 2018; Hayase, 2019); 3) a wide, randomly initialized MLP corresponds to the same Gaussian process whether or not its weights across layers are tied (Poole et al., 2016), and similarly, a simple RNN evolves the same whether or not its weights are tied across time, as long as its width is large (Chen et al., 2018). Indeed, we provide empirical evidence that calculations performed under this assumption still have considerable predictive power, even in cases where it is violated.

## 4.2 FORWARD SIGNAL PROPAGATION

We now study how correlations between inputs propagate into the hidden states of a recurrent network. This section follows closely the development in Chen et al. (2018). We now consider two sequences of normalized inputs $\{\mathbf{z}_a^t\}, \{\mathbf{z}_b^t\}$ with zero mean and covariance $\mathbf{R} = R \begin{pmatrix} 1 & \Sigma_z \\ \Sigma_z & 1 \end{pmatrix}$ fed into two copies of a network with identical weights, and resulting in sequences of states $\{\mathbf{s}_a^t\}, \{\mathbf{s}_b^t\}$. We consider the time evolution of the moments and correlations

$$\mu_s^t = \mathbb{E}[s_{ia}^t] \tag{2a}$$

$$Q_s^t \equiv Q_{s,aa}^t = \mathbb{E}[s_{ia}^t s_{ia}^t] \tag{2b}$$

$$\Sigma_s^{t2} C_s^t + (\mu_s^t)^2 = \mathbb{E}[s_{ia}^t s_{ib}^t] \tag{2c}$$

where we define $\Sigma_s^{t2} = Q_s^t - (\mu_s^t)^2$.

Returning to the pre-activations defined in eqn. (1). We make the *mean field approximation*[2] that the pre-activations are jointly Gaussian at the infinite width $N \to \infty$ limit:

$$\begin{pmatrix} u_{kia}^t \\ u_{kib}^t \end{pmatrix} \sim \mathcal{N}\left( \begin{pmatrix} \mu_k \\ \mu_k \end{pmatrix}, (Q_k^t - \mu_k^2) \begin{pmatrix} 1 & C_k^t \\ C_k^t & 1 \end{pmatrix} \right) \tag{3}$$

where the second moment $Q_k^t$ is given by

$$Q_k^t = \mathbb{E}[u_{kia}^t u_{kia}^t] = \sigma_k^2 \mathbb{E}[s_{ia}^t s_{ia}^t] + \nu_k^2 \mathbb{E}[z_{ia}^t z_{ia}^t] + \rho_k^2 + \mu_k^2 = \sigma_k^2 Q_s^t + \nu_k^2 R + \rho_k^2 + \mu_k^2 \tag{4a}$$

and defining $\Sigma_k^{t2} = Q_k^t - \mu_k^2$

$$C_k^t = \frac{\mathbb{E}[u_{kia}^t u_{kib}^t] - \mu_k^2}{Q_k^t - \mu_k^2} = \frac{\sigma_k^2 \left( \Sigma_s^{t2} C_s^t + \mu_s^2 \right) + \nu_k^2 R \Sigma_z + \rho_k^2}{Q_k^t - \mu_k^2}. \tag{4b}$$

The variables $u_{k_2 ia}^t, u_{k_2 ib}^t$ given by eqn. (1c) are also asymptotically Gaussian, with their covariance detailed in Appendix A. We will subsequently drop the vector subscript $i$ since all elements are

---

[2]For feed-forward networks, it has been shown using the Central Limit Theorem for exchangeable random variables that the pre-activations at all layers indeed converge in distribution to a multivariate Gaussian (Matthews et al. (2018)).

identically distributed and $\mathbf{f}, \mathbf{g}_k$ act element-wise, and the input sequence index in expressions that involve only one sequence.

For any $l \leq t$, the $u_k^l$ are independent of $s^l$ at the large $N$ limit. Combining this with the fact that their distribution is determined completely by $\mu_s^{l-1}, Q_s^{l-1}, C_s^{l-1}$ and that $\mathbf{f}$ is affine, one can rewrite eqn. (2a-c) using eqn. (1a-c) as the following deterministic dynamical system

$$(\mu_s^t, Q_s^t, C_s^t) = \mathcal{M}(\mu_s^{t-1}, Q_s^{t-1}, C_s^{t-1}, ..., \mu_s^1, Q_s^1, C_s^1) \tag{5}$$

where the dependence on $\Theta$ and the data distribution has been suppressed. In the peephole LSTM and GRU, the form will be greatly simplified to $\mathcal{M}(\mu_s^{t-1}, Q_s^{t-1}, C_s^{t-1})$. In Appendix E we compare the predicted dynamics to simulations, showing good agreement.

One can now study the fixed points of eqn. (5) and the rates of convergence by linearizing the dynamics. The fixed point of eqn. (5) is pathological, in the sense that any information that distinguishes two input sequences is lost. Therefore, delaying the convergence to the fixed point should allow for signals to propagate across longer time horizons. Quantitatively, **the rate of convergence of eqn. (5) to a fixed point gives an effective time scale for signal propagation from the inputs to the hidden state at later times**.

While the dynamical system is two-dimensional and analysis of convergence rates should be performed by linearizing the full system and studying the smallest eigenvalue of the resulting matrix, in practice as in Chen et al. (2018) this eigenvalue appears to always corresponds to the $C_s^t$ direction[3] . Hence, if we assume convergence of $Q_s^t, \mu_s^t$ we need only study

$$C_s^t = \mathcal{M}_C(\mu_s^*, Q_s^*, C_s^{t-1}) \tag{6}$$

where $\mathcal{M}_C$ also depends on expectations of functions of $\{\mathbf{u}_k^1\}, ...\{\mathbf{u}_k^{t-2}\}$ that do not depend on $C_s^{t-1}$. While this dependence is in principle on an infinite number of Gaussian variables as the dynamics approach the fixed point, $\mathcal{M}_C$ can still be reasonably approximated in the case of the LSTM as detailed in Appendix D. We show that in the case of the peephole LSTM this map is convex in Appendix C. This can be shown for the GRU by a similar argument. It follows directly that it has a single stable fixed point in these cases.

The rate of approach to the fixed point, $\chi_{C_s^*}$, is given by linearizing around it. Setting $C_s^t = C_s^* + \varepsilon^t$, we have

$$C_s^{t+1} = C_s^* + \varepsilon^{t+1} = \mathcal{M}_C(\mu_s^*, Q_s^*, C_s^* + \varepsilon^t) = \mathcal{M}_C(\mu_s^*, Q_s^*, C_s^*) + \chi_{C_s^*} \varepsilon^t + O((\varepsilon^t)^2). \tag{7}$$

The time scale of convergence to the fixed point is thus given by

$$\xi_{C_s^*} = -\frac{1}{\log \chi_{C_s^*}}, \tag{8}$$

which diverges as $\chi_{C_s^*}$ approaches 1 from below. Due to the detrimental effect of convergence to the fixed point described above, it stands to reason that a choice of $\Theta$ such that $\chi_{C_s^*} = 1 - \delta$ for some small $\delta > 0$ would enable signals to propagate from the initial inputs to the final hidden state when training on long sequences.

## 4.3 BACKWARDS SIGNAL PROPAGATION - THE STATE-TO-STATE JACOBIAN

We now turn to controlling the gradients of the network. A useful object to consider in this case is the asymptotic state-to-state transition Jacobian

$$\mathbf{J} = \lim_{t \to \infty} \frac{\partial \mathbf{s}^{t+1}}{\partial \mathbf{s}^t}.$$

$\mathbf{J}$ and powers of it will appear in the gradient as the covariance dynamics described in Section 4.2 approach their fixed point (specifically, the gradient of a network trained on a sequence of length $T$ will depend on a matrix polynomial of order $T$ in $\mathbf{J}$), hence we desire to control the squared singular

---

[3]This observation is explained in the case of fully-connected feed-forward networks in Schoenholz et al. (2016).

value distribution of $\mathbf{J}$. This distribution is described by moments of the form $m_{\mathbf{JJ}^T,n} = \tau((\mathbf{JJ}^T)^n)$, with mean and variance of the distribution given by

$$m_{\mathbf{JJ}^T,1} = \mathbb{E} \sum_{k \in K \cup \{0\}} a_k \tag{9}$$

$$\sigma^2_{\mathbf{JJ}^T} = m_{\mathbf{JJ}^T,2} - m^2_{\mathbf{JJ}^T,1} = \mathbb{E} \sum_{k,l \in K \cup \{0\}} 2a_k a_l - a_0^2 - \left( \mathbb{E} \sum_{k \in K \cup \{0\}} a_k \right)^2 \tag{10}$$

where the scalars $\{a_k\}_{k \in K \cup \{0\}}$ are architecture and hyperparameter dependent and are given in Appendix A.2.

Forward and backward signal propagation are in fact intimately related, as the following lemma shows:

**Lemma 1.** *For a recurrent neural networks defined by (1), the mean squared singular value of the state-to-state Jacobian defined in (16) and $\chi_{C_s}$ that determines the time scale of forward signal propagation (given by (7)) are related by*

$$m_{\mathbf{JJ}^T,1} = \chi_{C_s^*=1,\Sigma_z=1} \tag{11}$$

*Proof.* See Appendix C. $\qquad\square$

If one can find a setting of the hyperparameters such that $m_{\mathbf{JJ}^T,1}$ is close to $1$ and $\sigma^2_{\mathbf{JJ}^T}$ is small, then the spectrum of powers of $\mathbf{J}$ should remain well-conditioned ensuring that gradients will not explode or vanish.

## 4.4 DYNAMICAL ISOMETRY

Combining the results of Sections 4.2 and 4.3, we conclude that an effective choice of initialization hyperparamters should satisfy

$$\chi_{C_s^*} = 1 \tag{12a}$$

$$m_{\mathbf{JJ}^T,1} = 1 \tag{12b}$$

$$\sigma^2_{\mathbf{JJ}^T} = 0. \tag{12c}$$

We refer to these as *dynamical isometry* conditions. eqn. (12)a ensures stable signal propagation from the inputs to the loss, and eqn. (12)b-c are motivated by the additional requirement of preventing the gradients from exploding/vanishing. Both of these conditions appear to be necessary in order for a network to be trainable. Combining eqns. (9, 10, 11) we find that if $\Sigma_z = 1$, the dynamical isometry conditions are satisfied if $a_0 = 1, a_{k \neq 0} = 0$, which can be achieved by setting $\forall k : \sigma_k^2 = 0$ and taking $\mu_f \to \infty$. This motivates the general form of the initializations used in the experiments.

We demand that these equations are only satisfied approximately since for a given architecture, there may not be a value of $\Theta$ that satisfies them all. Additionally, even if such a value exists, the optimal value of $\chi_{C_s^*}$ for a given task may not be 1. There is some empirical evidence that if the characteristic time scale defined by $\chi_{C_s^*}$ is much larger than that required for a certain task, performance is degraded.

The typical values of $\Sigma_z$ will depend on the data set, yet satisfying the dynamical isometry conditions is simplified if $\Sigma_z = 1$ due to Lemma 1. It is thus a natural choice, yet we acknowledge that a more comprehensive treatment should consider the case of general $\Sigma_z$. We also find empirically that the results obtained under the $\Sigma_z = 1$ assumption prove to be predictive and enable training on a number of tasks without requiring a detailed analysis of $\Sigma_z$.

## 4.5 THE STATIONARY DISTRIBUTION OF THE LSTM CELL STATE

In architectures that were considered in previous works, as well as the peephole LSTM and GRU, all the quantities required to calculate the dynamical isometry conditions through eqns. (9, 10) can be written in terms of integrals over a finite number of Gaussian variables at the large width and time

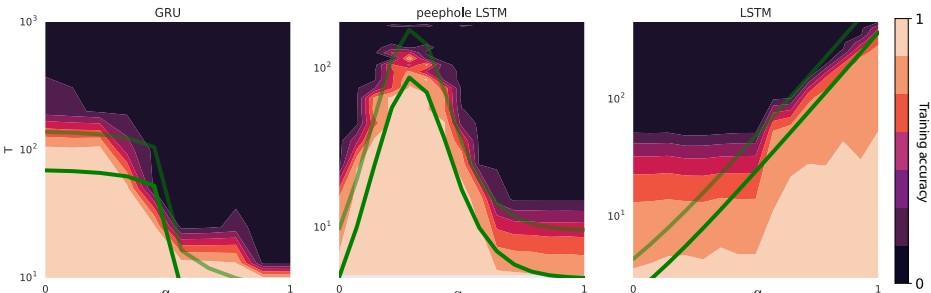

Figure 2: Training accuracy on the padded MNIST classification task described in 5.1 at different sequence lengths $T$ and hyperparameter values $\Theta_0 + \alpha\Theta_1$ for networks with untied weights, with different values of $\Theta_0, \Theta_1$ chosen for each architecture. The dark and light green curves are respectively $3\xi, 6\xi$ where $\xi$ is the theoretical signal propagation time scale in eqn. (8). As can be seen, this time scale predicts the transition between the regions of high and low accuracy across the different architectures and directions in hyperparameter space.

limit and are thus easy to evaluate. This is not the case in the standard LSTM due to the presence of the cell state. By unrolling the cell state update equation

$$\mathbf{c}^t = \sigma(\mathbf{u}_f^t) \circ \mathbf{c}^{t-1} + \sigma(\mathbf{u}_i^t) \circ \tanh(\mathbf{u}_r^t) \tag{13}$$

in time, we find that $\mathbf{c}^t$ depends on the entire sequence of $\mathbf{u}_k^t$ from the first time step [4].

From eqn. 13 we find that the asymptotic cell state distribution is that of a *perpetuity*, which is a random variable $X$ that obeys $X \overset{d}{=} XY + Z$ where $Y, Z$ are random variables and $\overset{d}{=}$ denotes equality in distribution. The stationary distribution of a perpetuity generally does not have a closed form, and neither does its likelihood (Goldie (1991)). In practice, one can overcome this difficulty by sampling from the stationary cell state distribution, as described in Appendix D.

## 5    EXPERIMENTS

### 5.1    PADDED MNIST CLASSIFICATION

The calculations presented above predict a characteristic time scale $\xi$ (defined in (8)) for forward signal propagation in a recurrent network. It follows that on a task where success depends on propagation of information from the first time step to the final $T$-th time step, the network will not be trainable for $T \gg \xi$. In order to test this prediction, we consider a classification task where the inputs are sequences consisting of a single MNIST digit followed by $T - 1$ steps of i.i.d Gaussian noise and the targets are the digit labels. By scanning across certain directions in hyperparameter space, the predicted value of $\xi$ changes. We plot training accuracy of a network trained with untied weights after 1000 iterations for the GRU and 2000 for the LSTM, as a function of $T$ and the hyperparameter values, and overlay this with multiples of $\xi$. As seen in Figure 2, we observe good agreement between the predicted time scale of signal propagation and the success of training. As expected, there are some deviations when training without enforcing untied weights, and we present the corresponding plots in the supplementary materials.

### 5.2    UNROLLED MNIST AND CIFAR-10

The calculation results motivate critical initializations that we test on standard long sequence benchmarks. The details of the initializations are presented in Appendix E. We unroll an MNIST digit into a sequence of length 784 and

|  | MNIST | CIFAR-10 |
|---|---|---|
| standard LSTM | 98.6 | 58.8 |
| critical LSTM | **98.9** | **61.8** |

Table 2: Test accuracy on unrolled MNIST and CIFAR-10.

train a critically initialized peephole LSTM with 600 hidden units. We also train a critically initialized

---

[4]In contrast, the hidden state distribution of the GRU at time $t$ for instance can be expressed as an integral over $\{\mathbf{u}_k^{t-1}\}$ alone (which are Gaussian at the large $t$ fixed point) through eqn. 2.

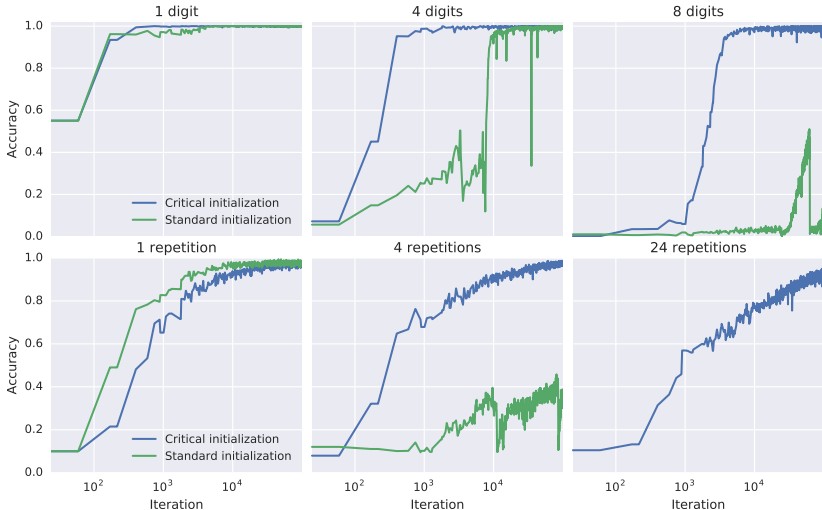

Figure 3: Training accuracy for unrolled, concatenated MNIST digits (*top*) and unrolled MNIST digits with replicated pixels (*bottom*) for different sequence lengths. *Left*: For shorter sequences the standard and critical initialization perform equivalently. *Middle*: As the sequence length is increased, training with a critical initialization is faster by orders of magnitude. *Right*: For very long sequence lengths, training with a standard initialization fails completely (and is unstable from initialization in the lower right panel).

LSTM with hard sigmoid nonlinearities on unrolled CIFAR-10 images feeding in 3 pixels at every time step, resulting in sequences of length 1024. We also apply standard data augmentation for this task. We present accuracy on the test set in Table 2. Interestingly, in the case of CIFAR-10 the best performance is achieved by an initialization with a forward propagation time scale $\xi$ that is much smaller than the sequence length, suggesting that information sufficient for successful classification may be obtained from a subset of the sequence.

### 5.3 Repeated pixel MNIST and multiple digit MNIST

In order to generate longer sequence tasks, we modify the unrolled MNIST task by repeating every pixel a certain number of times and set the input dimension to 7. To create a more challenging task, we also combine this pixel repetition with concatenation of multiple MNIST digits (either 0 or 1), and label such sequences by a product of the original labels. In this case, we set the input dimension to 112 and repeat each pixel 10 times. We train a peephole LSTM with both a critical initialization and a standard initialization on both of these tasks using SGD with momentum. In this former task, the dimension of the label space is constant (and not exponential in the number of digits like in the latter). In both tasks, we observe three distinct phases. If the sequence length is relatively short the critical and standard initialization perform equivalently. As the sequence length is increased, training with a critical initialization is faster by orders of magnitude compared to the standard initialization. As the sequence length is increased further, training with a standard initialization fails, while training with a critical initialization still succeeds. The results are shown in Figure 3.

## 6 Discussion

We have derived initialization schemes for recurrent networks motivated by ensuring stable signal propagation from the inputs to the loss and of gradient information from the loss to the weights. These schemes dramatically improve performance on long sequence tasks.

The subspace of initialization hyperparameters $\Theta$ that satisfy the dynamical isometry conditions is multidimensional, and there is no clear principled way to choose a preferred initialization within it. It would be of interest to study this subspace and perhaps identify preferred initializations based on additional constraints. One could also use the satisfaction of the dynamical isometry conditions as

a guiding principle in simplifying these architectures. A direct consequence of the analysis is that the forget gate, for instance, is critical, while some of the other gates or weights matrices can be removed while still satisfying the conditions. A related question is the optimal choice of the forward propagation time scale $\xi$ for a given task. As mentioned in Section 5.2, this scale can be much shorter than the sequence length. It would also be valuable understand better the extent to which the untied weights assumption is violated, since it appears that the violation is non-uniform in $\Theta$, and to relax the constant $\Sigma_z$ assumption by introducing a time dependence.

Another compelling issue is the persistence of the dynamical isometry conditions during training and their effect on the solution. It has been recently shown that in sufficiently wide networks the gradient flow dynamics in function space are effectively linear (Jacot et al. (2018); Lee et al. (2019)) and under these same conditions the dynamical isometry conditions persist during training. Understanding the finite width and learning rate corrections to such calculations could help extend the analysis of signal propagation at initialization to trained networks in this regime.

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

## Appendices

APPENDIX A    DETAILS OF RESULTS

A.1    COVARIANCES OF $\mathbf{u}_{k_2}^t$

The variables $u_{k_2 ia}^t, u_{k_2 ib}^t$ given by 1c are asymptotically Gaussian at the $N \to \infty$, with

$$Q_{k_2}^t = \sigma_{k_2}^2 \int g_k^2(u_a) \mathcal{D}_k Q_s^t + \nu_{k_2}^2 R + \rho_{k_2}^2 + \mu_{k_2}^2 \tag{14a}$$

$$C_{k_2}^t = \frac{\left( \begin{array}{c} \sigma_{k_2}^2 \int g_k(u_a) g_k(u_a) \mathcal{D}_k \left( \Sigma_s^{t2} C_s^t + \mu_s^2 \right) \\ + \nu_{k_2}^2 R \Sigma_z + \rho_{k_2}^2 \end{array} \right)}{Q_{k_2}^t - \mu_{k_2}^2} \tag{14b}$$

where $\mathcal{D}_k$ is a Gaussian measure on $(u_a, u_b)$ corresponding to the distribution in eqn. (3).

A.2    MOMENTS OF THE STATE-TO-STATE JACOBIAN

The moments of the squared singular value distribution are given by the normalized traces

$$m_{\mathbf{JJ}^T, n} = \tau((\mathbf{JJ}^T)^n).$$

Since $U^{t'}, t' < t$ is independent of $\mathbf{s}^t$, if we index by $k, k'$ the variables defined by eqn. (1b) and (1c) respectively we obtain

$$\mathbf{J} = \frac{\partial \mathbf{f}}{\partial \mathbf{s}^*} + \sum_k \frac{\partial \mathbf{f}}{\partial \mathbf{u}_k^*} \mathbf{W}_k + \sum_{k'} \frac{\partial \mathbf{f}}{\partial \mathbf{u}_{k_2'}^*} \mathbf{W}_{k_2'} \left( \mathbf{D}_{\mathbf{g}_k(\mathbf{u}_{k'}^*)} + \mathbf{W}_{k'} \mathbf{D}_{\mathbf{g}_{k'}'(\mathbf{u}_{k'}^*)} \mathbf{D}_{\mathbf{s}^*} \right). \tag{15}$$

Under the untied assumption $\mathbf{W}_k, \mathbf{W}_{k_2}$ are independent of $\mathbf{s}^t, \mathbf{u}_k^t$ at the large $N$ limit, and are also independent of each other and their elements have mean zero. Using this and the fact that $\mathbf{f}$ acts element-wise, we have

$$m_{\mathbf{JJ}^T, 1} = \tau(\mathbf{JJ}^T) = \mathbb{E} \sum_{k \in K \cup \{0\}} a_k \tag{16}$$

where

$$a_k = \begin{cases} D_0^2 & k = 0 \\ \sigma_k^2 D_k^2 & \mathbf{u}_k \text{ is given by (1b)} \\ \sigma_{k_2}^2 D_k^2 \left( \begin{array}{c} g_k^2(u_k^*) \\ + \sigma_k^2 s^{*2} g_k'^2(u_k^*) \end{array} \right) & \mathbf{u}_k \text{ is given by (1c)} \end{cases} \tag{17}$$

and $D_0 = \frac{\partial f}{\partial s^*}, D_k = \frac{\partial f}{\partial u_k^*}$. The values of $D_0, D_k$ for the architectures considered in the paper are detailed in Appendix A.3.

Controlling the first moment of $\mathbf{JJ}^T$ is not sufficient to ensure that the gradients do not explode or vanish, since the variance of the singular values may still be large. This variance is given by

$$\sigma_{\mathbf{JJ}^T}^2 = m_{\mathbf{JJ}^T, 2} - m_{\mathbf{JJ}^T, 1}^2.$$

The second moment $m_{\mathbf{JJ}^T,2}$ can be calculated from (15), and is given by

$$m_{\mathbf{JJ}^T,2} = \mathbb{E} \sum_{k,l \in K \cup \{0\}} 2a_k a_l - a_0^2 \tag{18}$$

where the $a_k$ are defined in eqn. (17). We arrange the scalars $a_0, \{a_k\}$ into a vector $\mathbf{a}$. In Appendix B we show that the results of the calculations in this section match the empirical spectrum of the Jacobian.

For all the architectures considered in this paper, we find that $D_0 = \sigma(u_f^*)$ while $D_k$ are finite as $\forall k : \sigma_k^2 \to 0$. Combining this with (11), (16), (18), we find that if $\Sigma_z = 1$ the dynamical isometry conditions are satisfied if $a_0 = 1, a_{k \neq 0} = 0$, which can be achieved by setting $\forall k : \sigma_k^2 = 0$ and taking $\mu_f \to \infty$. This motivates the general form of the initializations used in the experiments [5] although there are many other possible choices of $\Theta$ such that the $a_{k \neq 0}$ vanish.

Given the above general form of the dynamical isometry conditions for recurrent networks, we now provide the detailed forms that apply to the LSTM, peephole LSTM and GRU.

### A.3 DYNAMICAL ISOMETRY CONDITIONS FOR SELECTED ARCHITECTURES

We specify the form of $\chi_{C_s^*,\Sigma}$ and $\mathbf{a}$ for the architectures considered in this paper:

#### A.3.1 GRU

$$\chi_{C_s^*,\Sigma} = \mathbb{E} \left[ \begin{array}{c} \sigma(u_{fa}^*)\sigma(u_{fb}^*) + \sigma_f^2 \left( \begin{array}{c} \tanh(u_{r_2a}^*) \tanh(u_{r_2b}^*) \\ +h_a^* h_b^* \end{array} \right) \sigma'(u_{fa}^*)\sigma'(u_{fb}^*) \\ +\sigma_{r_2}^2(1 - \sigma(u_{fa}^*))(1 - \sigma(u_{fb}^*)) \tanh'(u_{r_2a}^*) \tanh'(u_{r_2b}^*) \left( \begin{array}{c} \sigma(u_{r_1a}^*)\sigma(u_{r_1b}^*) \\ +\sigma_{r_1}^2 h_a^* h_b^* \sigma'(u_{r_1a}^*)\sigma'(u_{r_1b}^*) \end{array} \right) \end{array} \right]$$

$$\mathbf{a} = \begin{pmatrix} \sigma^2(u_f^*) \\ \sigma_f^2 \left( \tanh^2(u_{r_2}^*) + Q_h^* \right) \sigma'^2(u_f^*) \\ \sigma_{r_1}^2 \sigma_{r_2}^2 h^{*2}(1 - \sigma(u_f^*))^2 \tanh'^2(u_{r_2}^*)\sigma'^2(u_{r_1}^*) \\ \sigma_{r_2}^2(1 - \sigma(u_f^*))^2 \tanh'^2(u_{r_2}^*)\sigma^2(u_{r_1}^*) \end{pmatrix}$$

#### A.3.2 PEEPHOLE LSTM

$$\chi_{C_s^*,\Sigma} = \mathbb{E} \left[ \begin{array}{c} \sigma(u_{fa}^*)\sigma(u_{fb}^*) + \sigma_i^2 \sigma'(u_{ia}^*)\sigma'(u_{ib}^*) \tanh(u_{ra}^*) \tanh(u_{rb}^*) \\ \sigma_f^2 c_a^* c_b^* \sigma'(u_{fa}^*)\sigma'(u_{fb}^*) + \sigma_r^2 \sigma(u_{ia}^*)\sigma(u_{ib}^*) \tanh'(u_{ra}^*) \tanh'(u_{rb}^*) \end{array} \right]$$

$$\mathbf{a} = \begin{pmatrix} \sigma^2(u_f^*) \\ \sigma_i^2 \sigma'^2(u_i^*) \tanh^2(u_r^*) \\ \sigma_f^2 c^{*2} \sigma'^2(u_f^*) \\ \sigma_r^2 \sigma^2(u_i^*) \tanh'^2(u_r^*) \end{pmatrix}$$

---

[5]In the case of the LSTM, we also want to prevent the output gate from taking very small values, as explained in Section A.3.

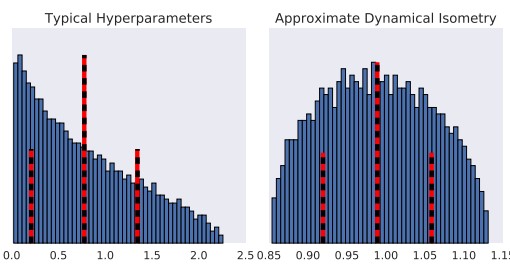

Figure 4: Squared singular values of the state-to-state Jacobian in eqn. (15) for two choices of hyperparameter settings $\Theta$. The red lines denote the empirical mean and standard deviations, while the dotted lines denote the theoretical prediction based on the calculation described in Section 4.3. Note the dramatic difference in the spectrum caused by choosing an initialization that approximately satisfies the dynamical isometry conditions.

### A.3.3 LSTM

$$\chi_{C_s^*,\Sigma_z} = \mathbb{E} \left[ \begin{array}{c} \sigma(u_{fa}^*)\sigma(u_{fb}^*) + \sigma_o^2 \sigma'(u_{oa}^*)\sigma'(u_{ob}^*)\tanh(c_a^*)\tanh(c_b^*) \\ +\sigma(u_{oa}^*)\sigma(u_{ob}^*)\tanh'(c_a^*)\tanh'(c_b^*) \left( \begin{array}{c} \sigma_f^2 c_a^* c_b^* \sigma'(u_{fa}^*)\sigma'(u_{fb}^*) \\ +\sigma_i^2 \sigma'(u_{ia}^*)\sigma'(u_{ia}^*)\tanh(u_{ra}^*)\tanh(u_{rb}^*) \\ +\sigma_r^2 \sigma(u_{ia}^*)\sigma(u_{ia}^*)\tanh'(u_{ra}^*)\tanh'(u_{rb}^*) \end{array} \right) \end{array} \right]$$

$$\mathbf{a} = \left( \begin{array}{c} \sigma^2(u_f^*) \\ \sigma_i^2 \sigma^2(u_o^*)\tanh'^2(c^*)\sigma'^2(u_i^*)\tanh^2(u_r^*) \\ \sigma_f^2 \sigma^2(u_o^*)\tanh'^2(c^*)c^{*2}\sigma'^2(u_f^*) \\ \sigma_r^2 \sigma^2(u_o^*)\tanh'^2(c^*)\sigma^2(u_i^*)\tanh'^2(u_r^*) \\ \sigma_o^2 \sigma'^2(u_o^*)\tanh^2(c^*) \end{array} \right)$$

When evaluating 7 in this case, we write the cell state as $c^t = \tanh^{-1}\left(\frac{s^t}{\sigma(o^t)}\right)$, and assume $\frac{\sigma(o^t)}{\sigma(o^{t-1})} \approx 1$ for large $t$. The stability of the first equation and the accuracy of the second approximation are improved if $o^t$ is not concentrated around 0.

### A.4 RNN TABLE SUPPLEMENT

|  | Minimal RNN Chen et al. (2018) | peephole LSTM Gers et al. (2002) |
|---|---|---|
| $K$ | $\{f, r\}$ | $\{i, f, r, o\}$ |
| $\mathbf{p}^t$ | $\mathbf{s}^t$ | $\sigma(\mathbf{u}_o^t) \circ \tanh(\mathbf{s}^t)$ |
| $\mathbf{f}$ | $\sigma(\mathbf{u}_f^t) \circ \mathbf{s}^{t-1}$ $+(\mathbf{1} - \sigma(\mathbf{u}_f^t)) \circ \mathbf{x}^t$ | $\sigma(\mathbf{u}_f^t) \circ \mathbf{s}^{t-1}$ $+\sigma(\mathbf{u}_i^t) \circ \tanh(\mathbf{u}_r^t)$ |

Table 3: Additional recurrent architectures written in the form 1. $\mathbf{p}^t$ is the output of the network at every time step. See Table 1 for more details.

## APPENDIX B SQUARED JACOBIAN SPECTRUM HISTOGRAMS

To verify the results of the calculation of the moments of the squared singular value distribution of the state-to-state Jacobian presented in Section 4.3 we run an untied peephole LSTM for 100 iterations with i.i.d. Gaussian inputs. We then compute the state-to-state Jacobian and calculate its spectrum. This can be used to compare the first two moments of the spectrum to the result of the calculation, as well as to observe the difference between a standard initialization and one close to satisfying the dynamical isometry conditions. The results are shown in Figure 4. The validity of this experiment rests on making an ergodicity assumption, since the calculated spectral properties require taking averages over realizations of random matrices, while in the experiment we instead calculate the moments by averaging over the

eigenvalues of a single realization. The good agreement between the prediction and the empirical average suggests that the assumption is valid.

## APPENDIX C   AUXILIARY LEMMAS AND PROOFS

**Proof of Lemma 1.** Despite the fact that each $u_{kai}$ as defined in 1 depends in principle upon the entire state vector $\mathbf{s}_a$, at the large N limit due to the isotropy of the input distribution we find that these random variables are i.i.d. and independent of the state. Combining this with the fact that $\mathbf{f}$ is an element-wise function, it suffices to analyse a single entry of $\mathbf{s}^t = \mathbf{f}(\mathbf{s}^{t-1}, \{\mathbf{u}_k^1\}, ..., \{\mathbf{u}_k^t\})$, which at the large $t$ limit gives

$$\mathcal{M}_C(\mu_s^*, Q_s^*, C_s) = \frac{\mathbb{E}\left[f(s_a, U_a^*)f(s_b, U_b^*)\right] - (\mu_s^*)^2}{Q_s^* - (\mu_s^*)^2}$$

where $U_a^* = \{u_{ka}^*|k \in K, u_{ka}^* \sim \mathcal{N}(\mu_k, Q_k^* - \mu_k^2)\}$ and $U_b^*$ is defined similarly (i.e. we assume the first two moments have converged but the correlations between the sequences have not, and in cases where $\mathbf{f}$ depends on a sequence of $\{\mathbf{u}_k^1\}, ..., \{\mathbf{u}_k^t\}$ we assume the constituent variables have all converged in this way). We represent $u_{ka}^*, u_{kb}^*$ via a Cholesky decomposition as

$$u_{ka}^* = \Sigma_k z_{ka} + \mu_k^* \tag{19a}$$

$$u_{kb}^* = \Sigma_k \left(C_k^* z_{ka} + \sqrt{1 - (C_k^*)^2} z_{kb}\right) + \mu_k^* \tag{19b}$$

where $z_{ka}, z_{kb} \sim \mathcal{N}(0,1)$ i.i.d. We thus have $\frac{\partial u_{kb}^*}{\partial C_k} = \sqrt{Q_k^* - (\mu_k^*)^2}\left(z_{ka} - \frac{C_k}{\sqrt{1-C_k^2}}z_{kb}\right)$. Combining this with the fact that $\int \mathcal{D}z g(z)z = \int \mathcal{D}z g'(z)$ for any $g(z)$, integration by parts gives for any $g_1, g_2$

$$\begin{aligned} &\frac{\partial}{\partial C_k} \int \mathcal{D}z_{ka}\mathcal{D}z_{kb}g_1(u_{ka}^*)g_2(u_{kb}^*) \\ &= \int \mathcal{D}z_{ka}\mathcal{D}z_{kb}g_1(u_{ka}^*)\frac{\partial g_2(u_{kb}^*)}{\partial u_{kb}^*}\frac{\partial u_{kb}^*}{\partial C_c} \\ &= \Sigma_k^2 \int \mathcal{D}z_{ka}\mathcal{D}z_{kb}\frac{\partial g_1(u_{ka}^*)}{\partial u_{ka}^*}\frac{\partial g_2(u_{kb}^*)}{\partial u_{kb}^*} \end{aligned} \tag{20}$$

Denoting $\Sigma_s^2 = Q_s^* - (\mu_s^*)^2$ and defining $\Sigma_k, \Sigma_{k_2}$ similarly, we have

$$\frac{dC_{ka}}{dC_s} = \frac{\sigma_{ka}^2 \Sigma_s^2}{\Sigma_{ka}^2} \tag{21}$$

$$\frac{dC_{kb}}{dC_s} = \frac{\sigma_{kb}^2 \Sigma_s^2}{\Sigma_{kb}^2}*$$
$$\int \left[\begin{array}{c} g_k(u_{ka}^*)g_k(u_{kb}^*) \\ +\frac{\sigma_{k_2}^2\left(\Sigma_s^2 C_s + (\mu_s^*)^2\right)}{\Sigma_{k_2}^2}\frac{\partial g_k(u_{ka}^*)g_k(u_{kb}^*)}{\partial C_k}\frac{dC_k}{dC_s} \end{array}\right] \mathcal{D}_k$$

$$\overset{\frac{\sigma_{k_2}^2 \Sigma_s^2}{\Sigma_{k_2}^2}*}{=} \int \left[ \begin{array}{c} g_k(u_{ka}^*)g_k(u_{kb}^*) \\ +\sigma_k^2 \left( \begin{array}{c} \Sigma_s^2 C_s \\ +\mu_s^{*2} \end{array} \right) g_k'(u_{ka}^*)g_k'(u_{kb}^*) \end{array} \right] \mathcal{D}_k$$

where in the last equality we used 20. Using 20 again gives

$$\frac{\partial \mathcal{M}_C(\mu_s^*, Q_s^*, C_s)}{\partial C_k} = \Sigma_k^2 \int \mathcal{D}z_{ka}\mathcal{D}z_{kb} \frac{\partial f(u_{ka}^*)}{\partial u_{ka}^*} \frac{\partial f(u_{kb}^*)}{\partial u_{kb}^*}$$

We now note, using 4, that if $C_s = 1, \Sigma_z = 1$ we obtain $C_k = 1$ and thus

$$\frac{dC_{k_2}}{dC_s}\bigg|_{C_s=1,\Sigma_z=1} = \frac{\sigma_{k_2}^2 \Sigma_s^2}{\Sigma_{k_2}^2} \left( \begin{array}{c} \int g_k^2(u_k^*)\mathcal{D}_k \\ +\sigma_k^2 Q_s^* \int (g_k'(u_k^*))^2 \mathcal{D}_k \end{array} \right)$$

$$\frac{\partial \mathcal{M}_C(\mu_s^*, Q_s^*, C_s)}{\partial C_k}\bigg|_{C_s=1,\Sigma_z=1} = \Sigma_k^2 \int \mathcal{D}z_k (\frac{\partial f(u_k^*)}{\partial u_k^*})^2$$

$$\frac{\partial \mathcal{M}_C(\mu_s^*, Q_s^*, C_s)}{\partial C_s}\bigg|_{C_s=1,\Sigma_z=1} = \mathbb{E}\left[ (\frac{\partial f(s)}{\partial s})^2 \right]$$

combining the above equations with 21 and comparing the result to 16 completes the proof.

$\square$

**Lemma 2.** *For any odd function $g(x)$,* $\left( \begin{array}{c} x \\ y \end{array} \right) \sim \mathcal{N}\left( \left( \begin{array}{c} \mu \\ \mu \end{array} \right), \Sigma \left( \begin{array}{cc} 1 & C \\ C & 1 \end{array} \right) \right), 0 \leq C \leq 1$ *we have* $\mathbb{E}g(x)g(y) \geq 0$.

**Proof of Lemma 2.** For $C = 1$ the proof is trivial. We now assume $0 \leq C < 1$. We split up $\mathbb{R}^2$ into four orthants and consider a point $(a, b)$ with $a, b \geq 0$. We have $g(a)g(b) = g(-a)g(-b) = -g(a)g(-b) = -g(-a)g(b) \geq 0$. We will show that $p(a, b) + p(-a, -b) > p(a, -b) + p(-a, b)$ where $p$ is the probability density function of $(x, y)$ and hence the points where the integrand is positive will contribute more to the integral than the ones where it is negative. Plugging these points into $p(x, y)$ gives

$$\frac{p(a,b) + p(-a,-b)}{p(a,-b) + p(-a,b)} = \frac{e^{-\alpha[(a-\mu)^2 - 2C(b-\mu)(a-\mu) + (b-\mu)^2]} + e^{-\alpha[(a+\mu)^2 - 2C(b+\mu)(a+\mu) + (b+\mu)^2]}}{e^{-\alpha[(a-\mu)^2 + 2C(b+\mu)(a-\mu) + (b+\mu)^2]} + e^{-\alpha[(a+\mu)^2 + 2C(b-\mu)(a+\mu) + (b-\mu)^2]}}$$

where $\alpha$ is some positive constant that depends on the determinant of the covariance (since $C < 1$ the matrix is invertible and the determinant is positive).

$$= \frac{\exp(2\alpha Cab)}{\exp(-2\alpha Cab)} \frac{\cosh(2\mu\alpha(1-C)(a+b))}{\cosh(2\mu\alpha(1-C)(a-b))} \geq 1$$

where the last inequality holds for $0 \leq C < 1$. It follows that the positive contribution to the integral is larger than the negative one, and repeating this argument for every $(a, b)$ in the positive orthant gives the desired claim (if $a = 0$ or $b = 0$ the four points in the analysis are not distinct but the inequality still holds and the integrand vanishes in any case). □

**Lemma 3.** *The map 5 is convex in the case of the peephole LSTM.*

**Proof of Lemma 3.** We have

$$\mathcal{M}(C_c^t) = \frac{\int f_a^t f_b^t \left((Q_c^* - (\mu_c^*)^2)C_c^t + \mu_c^{*2}\right) + \int i_a^t i_b^t \int r_a^t r_b^t + 2 \int f^t \int i^t \int r^t \mu_c^* - \mu_c^{*2}}{Q_c^* - \mu_c^{*2}}$$

.

From the definition of $u_{kb}^t$ and $C_k^t$ we have

$$\frac{\partial u_{kb}^t}{\partial C_c^t} = \sqrt{Q_k^* - (\mu_k^*)^2} \left(z_{ka} - \frac{C_k^t}{\sqrt{1 - C_k^{t2}}} z_{kb}\right) \frac{\partial C_k^t}{\partial C_c^t} = \left(z_{ka} - \frac{C_k^t}{\sqrt{1 - C_k^{t2}}} z_{kb}\right) \frac{\sigma_k^2(Q_c^* - \mu_c^{*2})}{\sqrt{Q_k^* - \mu_k^{*2}}}.$$

and using $\int \mathcal{D}x g(x)x = \int \mathcal{D}x g'(x)$ we then obtain for any $g(x)$

$$\begin{aligned} \frac{\partial}{\partial C_c^t} \int \mathcal{D}z_{ka} \mathcal{D}z_{kb} g(u_{ka}^t) g(u_{kb}^t) &= \int \mathcal{D}z_{ka} \mathcal{D}z_{kb} g(u_{ka}^t) g'(u_{kb}^t) \frac{\partial u_{kb}^t}{\partial C_c^t} \\ &= \sigma_k^2 (Q_c^* - \mu_c^{*2}) \int \mathcal{D}z_{ka} \mathcal{D}z_{kb} g'(u_{ka}^t) g'(u_{kb}^t) \end{aligned} \tag{22}$$

$$(Q_c^* - \mu_c^{*2})\frac{\partial^2 \mathcal{M}(C_c)}{\partial C_c^2} = \frac{\partial^2}{\partial C_c^2}\left[\int f_a f_b \left((Q_c^* - (\mu_c^*)^2)C_c + \mu_c^{*2}\right) + \int i_a i_b \int r_a r_b\right]$$

$$\begin{aligned} = &\frac{\partial^2 \int f_a f_b}{\partial C_c^2}\left((Q_c^* - (\mu_c^*)^2)C_c + \mu_c^{*2}\right) + 2\frac{\partial}{\partial C_c}\int f_a f_b(Q_c^* - (\mu_c^*)^2) \\ &+ \frac{\partial^2 \int i_a i_b}{\partial C_c^2}\int r_a r_b + \frac{\partial \int i_a i_b}{\partial C_c}\frac{\partial \int r_a r_b}{\partial C_c} + \int i_a i_b \frac{\partial^2 \int r_a r_b}{\partial C_c^2} \end{aligned}$$

From 22 and non-negativity of some of the integrands

$$\geq \frac{\partial^2 \int f_a f_b}{\partial C_c^2}\left((Q_c^* - (\mu_c^*)^2)C_c + \mu_c^{*2}\right) + \frac{\partial^2 \int i_a i_b}{\partial C_c^2}\int r_a r_b + \int i_a i_b \frac{\partial^2 \int r_a r_b}{\partial C_c^2}$$

.

From Lemma 2 we have $\int r_a r_b \geq 0$ and $\frac{\partial^2 \int g_a g_b}{\partial C_c^2} = \alpha \int g_a'' g_b'' \geq 0$ for $g = f, i, r$. We thus have

$$\frac{\partial^2 \mathcal{M}(C_c)}{\partial C_c^2} \geq 0$$

for $0 \leq C_c \leq 1$.

Convexity of this map has a number of consequences. One immediate one is that the map has at most one stable fixed point. □

## APPENDIX D    THE LSTM CELL STATE DISTRIBUTION

---

**Algorithm 1** LSTM hidden state moment fixed point iteration using cell state sampling

---

**function** FIXEDPOINTITERATION($\mu_s^{t-1}, Q_s^{t-1}, \Theta, n_s, n_{\text{iters}}$)
    $Q_k^{t-1} \leftarrow$ CALCULATEQK($Q_s^{t-1}, \Theta$)                      ▷ Using 4
    Initialize $\mathbf{c} \in \mathbb{R}^{n_s}$
    **for** $i \leftarrow 1$ to $n_{\text{iters}}$ **do**
        $\mathbf{u}_i, \mathbf{u}_f, \mathbf{u}_r \leftarrow$ SAMPLEUS($Q_k^{t-1}, \Theta$)             ▷ Using 3
        $\mathbf{c} \leftarrow$ UPDATEC($\mathbf{c}, \mathbf{u}_i, \mathbf{u}_f, \mathbf{u}_r$)                ▷ Using 13
    **end for**
    $(\mu_s^t, Q_s^t) \leftarrow$ CALCULATEM($\mu_s^{t-1}, Q_s^{t-1}, \Theta, \mathbf{c}$)          ▷ Using 5
    **return** $(\mu_s^t, Q_s^t)$
**end function**

---

As mentioned in the main text, the cell state differs substantially from other random variables that appear in this analysis since it cannot be expressed as a function of a finite number of variables that are Gaussian at the large $N$ and $t$ limit (see Table 1). Since at this limit the $\mathbf{u}_i^t$ are independent, by examining the cell state update eqn. (13) we find that the asymptotic cell state distribution is that of a perpetuity, which obeys

$$X \stackrel{d}{=} XY + Z$$

for some random variables $Y, Z$. Its stationary distributions will have heavy tails Goldie (1991). Due to the bounds on $Y, Z$ in the case of the LSTM, one expects based on results in (Goldie & Grübel (1996)) that the tails of the stationary cell state distribution will be exponential in the case of hard sigmoid nonlinearities or decay like $x^{-x}$ in the case of soft sigmoid nonlinearities, and this tail behavior is indeed observed in simulations. Aside from the tails, the bulk of the distribution can take a variety of different forms and can be highly multimodal, depending on the choice of $\Theta$ which in turn determines the distributions of $Y, Z$.

We overcome this difficulty by sampling from the stationary cell state distribution. For a given value of $Q_h$, the variables $\mathbf{u}_k^t$ appearing in (13) can be sampled since their distribution is given by (3) at the large $N$ limit. The update equation (13) can then be iterated and the resulting samples approximate well the stationary cell state distribution for a range of different choices of $\Theta$, which result in a variety of stationary distribution profiles (see Appendix E3). The fixed points of (5) can then be calculated numerically as in the deterministic cases, yet care must be taken since the sampling introduces stochasticity into the process.

An example of the fixed point iteration eqn. (5) implemented using sampling is presented in Algorithm 1. The correlations between the hidden states can be calculated in a similar fashion. In practice, once the number of samples $n_s$ and sampling iterations $n_{\text{iters}}$ is of order 100 reasonably accurate values for the moment evolution and the convergence rates to the fixed point are obtained (see for instance the right panel of Figure 2). The computational cost of the sampling is linear in both $n_s, n_{\text{iters}}$ (as opposed to say simulating a neural network directly in which case the cost is quadratic in $n_s$).

## APPENDIX E    ADDITIONAL EXPERIMENTS AND DETAILS OF EXPERIMENTS

### E.1    DYNAMICAL SYSTEM

We simulate the dynamics of $Q_s^t$ in eqn. (5) for a GRU using inputs with $\Sigma_z^t = 0$ for $t < 10$ and $\Sigma_z^t = 1$ for $t \geq 10$. Note that this evolution is independent of the evolution of $C_s^t$. The results show good agreement in the untied case between the calculation at the large $N$ limit and the simulation, as shown in Figure 5.

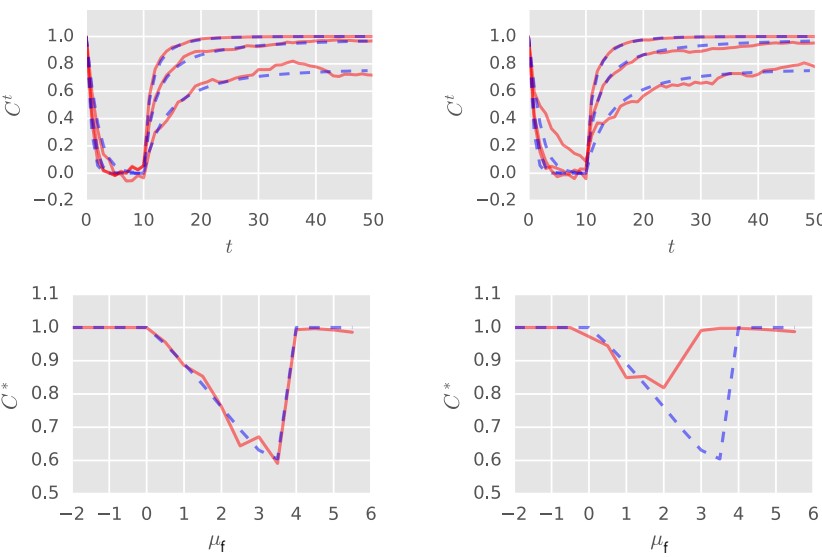

Figure 5: *Top*: Dynamics of the correlations (6) for the GRU with 3 different values of $\mu_f$ as a function of time. The dashed line is the prediction from the mean field calculation, while the red curves are from a simulation of the network with i.i.d. Gaussian inputs. *Left*: Network with untied weights. *Right*: Network with tied weights. *Bottom*: The predicted fixed point of (6) as a function of different $\mu_f$. *Left*: Network with untied weights. *Right*: Network with tied weights.

### E.2    HEATMAPS

In Figure 6 we present results of training on the same task shown in Figure 2 with tied weights, showing the deviations resulting from the violation of the untied weights assumption.

### E.3    SAMPLING THE LSTM CELL STATE DISTRIBUTION

As described in Appendix D, calculating the signal propagation time scale and the moments of the state-to-state Jacobian for the LSTM requires integrating with respect to the stationary cell state distribution. The method for doing this is described in Algorithm 1. As is shown in Figure 7, this distribution can take different forms based on the choice of initialization hyperparameters $\Theta$, but in all cases we have studied the proposed algorithm appears to provide a reasonable approximation to this distribution efficiently. The simulations are obtained by feeding a network of width $N = 200$ with i.i.d. Gaussian inputs.

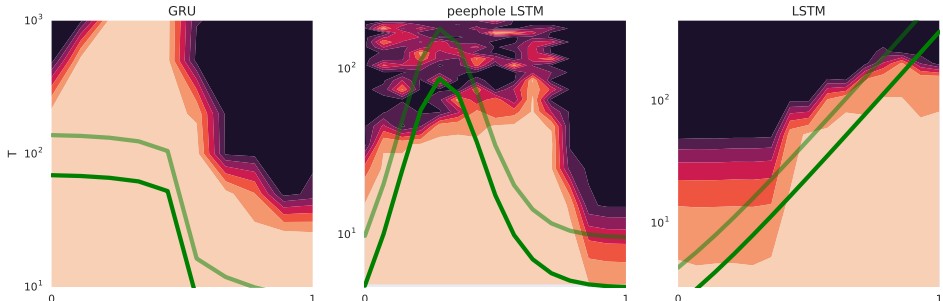

Figure 6: Training accuracy on the padded MNIST classification task described in 5.1 at different sequence lengths $T$ and hyperparameter values $\Theta$ for networks with *tied* weights. The green curves are multiples of the forward propagation time scale $\xi$ calculated under the *untied* assumption. We generally observe improved performance when the predicted value of $\xi$ is high, yet the behavior of the network with tied weights is not part of the scope of the current analysis and deviations from the prediction are indeed observed.

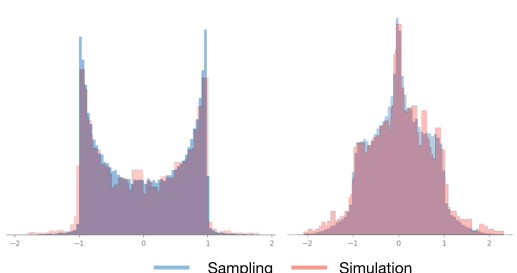

Figure 7: Sampling from the LSTM cell state distribution using Algorithm 1, showing good agreement with the cell state distribution obtained by simulating a network with untied weights. The two panels correspond to two different choices of $\Theta$

### E.4 CRITICAL INITIALIZATIONS

Peephole LSTM:

$$\mu_i, \mu_r, \mu_o, \rho_i^2, \rho_f^2, \rho_r^2, \rho_o^2, \nu_i^2, \nu_f^2, \nu_r^2, \nu_o^2 = 0$$
$$\mu_f = 5$$
$$\sigma_i^2, \sigma_f^2, \sigma_r^2, \sigma_o^2 = 10^{-5}$$

LSTM (Unrolled CIFAR-10 task):

$$\mu_i, \mu_r, \mu_o, \rho_i^2, \rho_f^2, \rho_r^2, \rho_o^2, \nu_f^2, \nu_o^2 = 0$$
$$\nu_i^2, \nu_r^2, \sigma_o^2, \mu_f = 1$$
$$\sigma_i^2, \sigma_f^2, \sigma_r^2 = 10^{-5}$$

The value of $\mu_f$ was found by a grid search, since for this task information necessary to solve it did not require signal propagation across the entire sequence. In other words, classification of an image can be achieved with access only to the last few rows of pixels. The utility of the analytical results in this case, as mentioned in the text, is to greatly constrain the hyperparameter space of potentially useful initializations from theoretical considerations.

### E.5 STANDARD INITIALIZATION

LSTM and peephole LSTM:

Kernel matrices (corresponding to the choice of $\nu_k^2$) : Glorot uniform initialization Glorot & Bengio (2010)

Recurrent matrices (corresponding to the choice of $\sigma_k^2$): Orthogonal initialization (i.i.d. Gaussian initialization with variance $1/N$ also used giving analogous performance)

$$\mu_i, \mu_r, \mu_o, \rho_i^2, \rho_f^2, \rho_r^2, \rho_o^2 = 0$$
$$\mu_f = 1$$

#### E.5.1 LONG SEQUENCE TASKS

Learning rate scan: 8 equally spaced points between $10^{-2}$ and $10^{-5}$. Validation set: 10000 images for MNIST and CIFAR-10. In Table 2, the results for the standard LSTM on MNIST were reproduced from Arpit et al. (2018) and the results on CIFAR-10 were reproduced from Trinh et al. (2018).

