# OpenReview forum: "The Dynamics of Signal Propagation in Gated Recurrent Neural Networks"
_ICLR.cc/2020/Conference — Reject_

### Official Review · AnonReviewer1 · 2019-10-21
**Official Blind Review #1**

**Rating:** 3

**Review:**

This paper touches the signal processing/long term propagation problem in gated recurrent neural networks from the mean field theory. The paper starts from a dynamic system view of the recurrent neural networks and calculates the time scale of converging to the fixed point. In order to avoid the system to converge to the fixed point, the authors utilize some initialization strategy to keep the time scale to infinity. The authors also relate the time scale to state-to-state Jacobians.

The paper is interesting but more details could be added to both theories and experiments. Given all those extra details, I can increase my scores.

For the theory:
1. It is unclear how you go from equation (7) to (8). References or more explanations need to be added.
2. It is unclear how the initializations in E.4 satisfies the criteria you define in the paper. More explanations can be added.
3. This mean field approximation is still far away from practice. It is hard to believe the input in real life is Gaussian distributed vectors (sec 4.2). What will happen if the input distributions are not Gaussian? This should be discussed.
4. This initialization only helps at the beginning of the training. What will happen if we do one backpropagation? This should be addressed.

For the experiments:
1. The authors did not state the some of the experiment details in the papers, like what optimizer, regularization, learning rate.... To make better assessment of the experiments, those details should be added. Do you train/tune all the different initializations in the same way?
2. I cannot find the description of Figure 1 anywhere in the paper. It is hard to believe LSTM did poorly on sequential MNIST unless giving more details since LSTM has been proved to perform okay on sequential MNIST in a bunch of papers[1].
3. What is the meaning of 112 dimension in 5.3? Does that mean you only choose the first 4 rows of MNIST images?
4. Comparison over random seeds should be honestly justified for all the experiments.

Minors:
1. You have one missing \Sigma_z in equation (4a).
2. \mu_s^2 in equation (4b) should be (\mu_s^t)^2

References:
[1] Arjovsky, Martin, Amar Shah, and Yoshua Bengio. "Unitary evolution recurrent neural networks." International Conference on Machine Learning. 2016.

-------------
I change my scores to weak reject . I agree with Reviewer 2 that the paper provide some insight from Physics and can be an interesting contribution to the community. However, I think all of the reviewers. including myself, find the paper hard to read for the general machine learning audience. And even though the authors mention that they will fix the text in the future, they do not change any text of the paper. I think writing is also important besides presenting interesting research ideas. Overall, I think the paper will be benefited from resubmission.

**Experience Assessment:**

I have read many papers in this area.

**Review Assessment: Checking Correctness Of Derivations And Theory:**

I carefully checked the derivations and theory.

**Review Assessment: Checking Correctness Of Experiments:**

I carefully checked the experiments.

**Review Assessment: Thoroughness In Paper Reading:**

I read the paper thoroughly.

---

### Official Review · AnonReviewer2 · 2019-10-22
**Official Blind Review #2**

**Rating:** 8

**Review:**

The aim of this paper is to suggest randomized initializations for the various weights of a recurrent neural network (GRUs and various LSTMs are covered), such that training these networks gets to a successful start, when the model is trained on long sequences. Instead of being heuristic, their approach follows first principles of analyzing signal propagation through time, using ideas from statistical thermodynamics (mean field approximations). Some experiments, on toy datasets, validate their approach.

I am quite intrigued by this paper. It is using interesting theory, shapes it to a practically highly relevant and difficult applied problem, and in the end comes up with a computable criterion of how to choose hyperparameters (means and variances of Gaussians to sample initial weights from). While the results in practice are still not too convincing, I am strongly in favour of giving this approach the benefit of doubt, as it could lead to practically very useful downstream work.

The main direction of improvement for this paper (given that experiments are what they are -- somewhat limited to toy situations right now) is to better explain the methodology to researchers not familiar with mean field methods. Most importantly, it is not explained in the main text how hyperparameters are really chosen in the end. Looking at Appendix E, I find some pretty basic choices, and no other alternatives considered. It is not explained why these choices satisfy the theory, why they'd be the only ones, etc. This creates a disconnect between the very nice (and seemingly useful) theory and its implications (they are not really well spelled out).

Here is what I understood (and I am not specifically an expert on stat mech). The authors assume that the dimension of latent states (N) grows large. They assume that weights are sampled independently, and identical distributed in groups k (different cell types, weight vs bias), and that inputs are correlated with each other in each dimension. Based on these assumption, they follow Gaussians statistics through a number of time steps. In the limit, one gets a deterministic dynamical system, and as t -> infty, this may converge to a fixed point. In a very nice argument (which they could explain better), they state that such rapid convergence is bad news, because then information cannot spread across long time scales, so one has to find hyperparameters for which the system behaves "critically". A second arguments tries to keep gradient sizes (under MF assumptions) of O(1), so neither -> 0, not -> infty, which is again some "critical" range. Under their assumption, these critical conditions can be computed depending on the hyperparameters.

Unfortunately, this is where the paper somewhat stops, it does not give specific methods for finding hyperpars that satisfy the criteria, at least not in the sense of characterising the whole space of such hyperpars (instead, in Appendix E, they just state some few settings that do). As a direction for future work, this would be very important. Another side question is whether for what the authors call "trainability", the only point that matters is whether for the initial weights, signals can spread and gradients are O(1). It is certainly necessary, I see that.

Detailed comments:
- Please fix Table 1, the expressions seem broken. What does "r2" mean in the GRU column?
- At least for me, (1a) to (1c) really was too short. At least in the Appendix, please do explain how this gives GRU and LSTM
- Please explain the untied weight assumption somewhere. s^t is a map of s^(t-1) and W_k, so how can W_k be independent
   of all s^t? What are you really assuming here?
- It took some repeated reading until I understood why the expressions in (2a) to (4b) do not depend on i, j, a, b (except
  whether a = b or a != b). Explain that properly
- The core of the whole approach seems to be first half of page 5. This seems like a very nice argument, but hard to understand. Try making it more crisp. I kind of get the rough idea why fast convergence over t would be bad, but would total divergence over t not also be bad?
- In (12a-c), do you mean "equal" or "approximately equal"?
- In 4.4: "This motivates the general form of the initializations used in the experiments": You have to make this more explicit. Why are your choices the only ones? Could there not be other choices satisfying (12a-c) approx, and be better?
- Value of Sigma_z = 1: This seems odd to me, then your covariances are degenerate (rank 1 instead of 2). Please explain
- Standard LSTM harder than GRU or peephole LSTM: Again, this sounds real interesting, but I did not get it from the explanation
- I did not understand Figure 2. How are Theta_0, Theta_1 chosen?
- As said above, the experiments are interesting, but somewhat artificial. Please do at least comment on real-world applications, and whether (and how) the ideas here would apply
- Discussion: "there is no clear principled way...": Well, but practitioners need something. I'd disagree, at least one could attempt to navigate this space by global optimization techniques...


ADDITIONAL COMMENT:

I tried to append the following as comment, but the (pretty broken) system would not let me, insisting that "reader is not valid" (???). Anyway, here it is. I hope I am allowed to add to my own review.

I've seen the argument in reviews that assumptions made by this paper about independencies between weights and inner states are wrong, and therefore conclusions are not valid.

First, such assumptions are indeed pretty common in such statistical mech analyses of learning methods. Second, you have to distinguish between weights after (random) initialization and after training. Of course, LSTM represents long term dependencies after training, but initialization is a different story.

If I was the AC for this paper, I'd ask somebody with at least some background in statistical mech to provide some additional opinions, as the reviewers (including myself) are not fully qualified.


**Experience Assessment:**

I have published in this field for several years.

**Review Assessment: Checking Correctness Of Derivations And Theory:**

I assessed the sensibility of the derivations and theory.

**Review Assessment: Checking Correctness Of Experiments:**

I did not assess the experiments.

**Review Assessment: Thoroughness In Paper Reading:**

I read the paper thoroughly.

---

### Official Review · AnonReviewer3 · 2019-10-23
**Official Blind Review #3**

**Rating:** 1

**Review:**

The authors propose a mean-field analysis of recurrent networks in this paper. I have a few concerns about this paper:

(1) The most serious concern about their analysis comes from their assumption. They assume the weight W is independent on the state s_t (Page 4, Lines 5-6). The recursive structure is the most complicated part of the recurrent networks, and also its major difference from feedforward networks. In current networks, the hidden states become (or even heavily) dependent on the weight due to recursion.

When making such an assumption, the recurrent networks just become similar to feedforward networks. The authors' claim that "the untied weights assumption actually has long history of yielding correct prediction" is not ungrounded and questionable.

(2) The paper is not well written. Some assumptions are not explicitly stated. They are placed in the text without any highlight. Some theoretical statements are claimed without any rigorous proof. A few approximations are applied without clearly explaining about the resulting approximation errors. This is not acceptable, especially when the authors claim they are developing a "THEORY".

(3) The experiments only consider the MNIST and CIFAR10 datasets. These datasets are mainly used for evaluating feedforward-type convolutional neural networks. Even though the authors might like their experiments, for the sake of the main stream users of recurrent networks. They should at least include experiments in conventional sequential prediction problems, e.g., speech, time series, machine translations.

(4) Compared with other state of the art methods, their experimental results on CIFAR10 is too weak. I cannot believe such weak results can be used to make meaningful justifications.










**Experience Assessment:**

I have published in this field for several years.

**Review Assessment: Checking Correctness Of Derivations And Theory:**

I carefully checked the derivations and theory.

**Review Assessment: Checking Correctness Of Experiments:**

I carefully checked the experiments.

**Review Assessment: Thoroughness In Paper Reading:**

I read the paper thoroughly.

---

### Decision · Program_Chairs · 2019-12-19

**Decision:**

Reject

**Comment:**

Using ideas from mean-field theory and statistical mechanics, this paper derives a principled way to analyze signal propagation through gated recurrent networks.  This analysis then allows for the development of a novel initialization scheme capable of mitigating subsequent training instabilities.  In the end, while reviewers appreciated some of the analytical insights provided, two still voted for rejection while one chose accept after the rebuttal and discussion period.  And as AC for this paper, I did not find sufficient evidence to overturn the reviewer majority for two primary reasons.

First, the paper claims to demonstrate the efficacy of the proposed initialization scheme on multiple sequence tasks, but the presented experiments do not really involve representative testing scenarios as pointed out by reviewers.  Given that this is not a purely theoretical paper, but rather one suggesting practically-relevant initializations for RNNs, it seems important to actually demonstrate this on sequence data people in the community actually care about.  In fact, even the reviewer who voted for acceptance conceded that the presented results were not too convincing (basically limited to toy situations involving Cifar10 and MNIST data).

Secondly, all reviewers found parts of the paper difficult to digest, and while a future revision has been promised to provide clarity, no text was actually changed making updated evaluations problematic.  Note that the rebuttal mentions that the paper is written in a style that is common in the physics literature, and this appears to be a large part of the problem.  ICLR is an ML conference and in this respect, to the extent possible it is important to frame relevant papers in an accessible way such that a broader segment of this community can benefit from the key message.  At the very least, this will ensure that the reviewer pool is more equipped to properly appreciate the contribution.  My own view is that this work can be reframed in such a way that it could be successfully submitted to another ML conference in the future.